# Arrest of Cell Cycle by Avian Reovirus p17 through Its Interaction with Bub3

**DOI:** 10.3390/v14112385

**Published:** 2022-10-28

**Authors:** Junyu Tang, Mengjiao Fu, Xiang Chen, Yimeng Zhao, Li Gao, Hong Cao, Xiaoqi Li, Shijun J. Zheng, Yongqiang Wang

**Affiliations:** 1Key Laboratory of Animal Epidemiology of the Ministry of Agriculture, China Agricultural University, Beijing 100193, China; 2College of Veterinary Medicine, China Agricultural University, Beijing 100193, China

**Keywords:** avian reovirus, p17, cell-cycle arrest, Bub3, mitotic checkpoint, viral replication

## Abstract

Avian reoviruses (ARV) are a group of poultry pathogens that cause runting and stunting syndrome (RSS), a condition otherwise known as “frozen chicken”, which are characterized by dramatically delayed growth in broilers. It has been known that p17, a nonstructural protein encoded by ARV, prohibits cellular proliferation by halting the cell cycle at the G2/M phase, the result of which is directly associated with the typical clinical sign of RSS. Nevertheless, the mechanism by which p17 modulates cell-cycle progression remains largely unknown. Here, we screened the interactome of ectopically expressed p17 through a yeast two-hybrid assay and identified Bub3, a cellular mitotic checkpoint protein, as a binding partner of p17. The infection of the Vero cells by ARV downregulated the Bub3 expression, while the knockdown of Bub3 alleviated the p17-modulated cell-cycle arrest during ARV infection. Remarkably, the suppression of Bub3 by RNAi in the Vero cells significantly reduced the viral mRNA and protein abundance, which eventually led to diminished virus replication. Altogether, our findings reveal that ARV p17 impedes host cell proliferation through a Bub3-dependent cell-cycle arrest, which eventually contributes to efficient virus replication. These results also unveil a hitherto unknown therapeutic target for RSS.

## 1. Introduction

Runting and stunting syndrome (RSS, also known as malabsorption syndrome) is an infectious enteric disease that mainly affects chickens and can lead to severe economic losses due to decreased weight gain, low feed conversion, abnormal feather development, high morbidity, etc., [1,2,3]. First reports of RSS outbreaks among poultry flocks can be traced back to the 1970s [4]. The causative agent of RSS is avian reovirus (ARV), a member of the reoviridae family that mainly infects poultry [5,6,7,8,9,10]. However, the basis of ARV pathogenesis and its link with RSS pathology have remained elusive.

The ARV genome consists of 10 double-stranded RNA segments wrapped by a double-layered nucleocapsid, which can be classified into three groups including large (L1−L3), medium (M1−M3) and small (S1−S4). ARV genomes mainly encode 8 structural proteins (λA, λB, λC, μA, μB, σA, σB, σC) and 4 nonstructural proteins (μNS, p10, p17, σNS) [11,12,13]. As a nucleoplasmic shuttling protein, p17, encoded by the S1 segment is known to interfere with gene transcription and autophagosome activation [11]. Importantly, p17 is also involved in the modulation of the cell cycle in the host cells via a complex and versatile mechanism, including the activation of p53 and p21^cip1/waf1^, and the inactivation of PI3K/AKT/mTOR and ERK signaling pathways [14,15,16].

The eukaryotic cell cycle is typically separated into four phases: the DNA synthesis phase (S), the mitosis phase (M), and two gap phases (G1 and G2). Upon receiving the growth signal, the quiescent cells maintained in a nondividing state enter the G1 phase, which prepare the cells for DNA synthesis and the active division. Next, the cells in the G1 phase are transits to the S phase, which is marked by robust DNA replication and chromosomal duplication. After DNA replication is completed, the G2 phase is initiated for cell division. The checkpoint delay of the G2 phase also ensures the absence of accumulated DNA damage or incomplete genome replication has occurred [17]. Once the mitosis is complete, the cells then enter the M phase [18,19]. Notably, the spindle assembly checkpoint (SAC, also known as mitotic checkpoint), a safeguard network that prevents inaccurate chromosome segregation, exists after the metaphase and prior to the anaphase. In order to avoid whole-chromosome gains or losses and their detrimental effects on cell physiology, SAC is activated when kinetochores fail to attach correctly to the microtubule spindle apparatus during the early stages of the prometaphase, which in turn catalyzes the formation of the mitotic checkpoint complex (MCC) assembled from the interaction of Mad2, Bub3, BubR1 and Cdc20 [20,21,22], leading to the inhibition of downstream anaphase promoting complex (APC/C) and block of cell-cycle progression [23,24,25]. As a crucial checkpoint protein, Bub3 has been proposed for contributing directly to the MCC generation by promoting binding of BubR1 to Cdc20. Apart from that, it was also reported that Bub3 activates APC/C^Cdc20^ at the kinetochore to promote anaphase onset. Only in the presence of unattached kinetochores, Bub3 functions in the inhibitory mitotic checkpoint complex to prevent APC/C^Cdc20^ substrate ubiquitination, allowing additional time to fix errors in attachment [26,27,28].

For achieving establishment and propagation, a number of different viruses have evolved to exploit various cellular resources to facilitate their own replication, including the manipulation of the cell cycle, examples of which include viral proteins that target cyclin-dependent kinase complexes or mitotic checkpoint proteins [17,29]. Recent studies showed that the expression of ARV p17 resulted in rapid cell-cycle arrest [15,16,30]. Unfortunately, the intrinsic mechanism by which p17 modulates the cell cycle remains largely unknown. In this study, we aimed to explore strategies employed by p17 to retard cell proliferation. As a consequence, we discovered a new approach, by which p17 functions to hamper cell-cycle progression.

## 2. Materials and Methods

### 2.1. Cell Cultures and Viruses

DF-1 cells, HEK 293T cells and Vero-E6 cells were purchased from ATCC. Cells were cultured in Dulbecco’s modified Eagle medium (DMEM) (Invitrogen, Carlsbad, CA, USA) supplemented with 10% fetal bovine serum (FBS) (Gibco, Grand Island, NE, USA) and penicillin/streptomycin (Macgene, Beijing, China) in a 5% CO_2_ incubator at 37 °C. The S1133 strain of ARV was kindly provided by Dr. Jingliang Su (China Agricultural University, Beijing, China).

### 2.2. Antibodies and Reagents

Sigma B mouse monoclonal antibodies were generated in our lab [31]. Anti-Myc antibody (2276) was obtained from Cell Signaling Technology (Boston, MA, USA). Anti-Flag antibody (F1804) was obtained from Merck Sigma Aldrich (St. Louis, MO, USA). Anti-Bub3 (133699) rabbit antibodies were obtained from Abcam (Cambridge, UK). Anti-GAPDH (60004) and anti-GFP (66002) mouse antibody were obtained from Proteintech (Wuhan, China). Anti-α-Tubulin (PM054) rabbit antibody was obtained from Medical Biological Laboratories (Tokyo, Japan). Horseradish peroxidase (HRP)-conjugated goat anti-mouse and anti-rabbit IgG antibodies were purchased from Ding Guo Sheng Wu (Beijing, China). Lipofectamine 3000 reagent was obtained from Invitrogen (Carlsbad, CA, USA). Cell cycle and apoptosis analysis kits were purchased from Beyotime Biotechnology (Shanghai, China).

### 2.3. Constructions of Recombinant Plasmids

The CDS of ARV p17 (GenBank accession ID: AF330703) was cloned from the genome of ARV S1133 with specific primers (sense primer, 5-ATGCAATGGCTCCGCCAT-3′; antisense primer, 5′-TCATAGATCGGCGTCAAATC-3′). The chicken Bub3 gene (GenBank ID: NM_001006506) was originally cloned from cDNA extracted from DF-1 cells using specific primers (sense primer, 5′-ATGAAAACGATGACGGGATC-3′; antisense primer, 5′-CTAAGTAGACTTAGGTTTTGTTTCTG-3′). PRK5-Flag-p17, pMRFP-p17, pEGFP-p17, pCMV-Myc-Bub3, pEGFP-Bub3 and pRK5-Flag-Bub3 were constructed by standard molecular biology techniques. All primers were synthesized by Sangon Company (Shanghai, China).

### 2.4. RNA Extraction and RT-qPCR

Total RNAs were prepared from Vero-E6 cells using an EASYspin Plus kit (Aidlab, Beijing, China), per the manufacturer’s instructions. Quantitative reverse transcription-PCR (qRT-PCR) was performed using a PrimeScript RT reagent kit (TaKaRa, Kusatsu, Japan) on a Light Cycler 480 II (Roche, Basel, Switzerland). Primers for ARV Sigma B were as follows: sense primer 5′-CGCTCCCTCTGCGATTAC-3′, antisense primer 5′- TCGTCGGCTCTGGTTTGA-3′. Primers for Bub3 were as follows: sense primer 5′-ATTGAGCAGATTTACCCAG-3′, antisense primer 5′-GGTGGTTCTGATGGCTT-3′. Primers for GAPDH were as follows: sense primer 5′- CTCATGACCACAGTCCATGC -3′, antisense primer 5′- CCCTGTTGCTGTAGCCAAAT -3′. The primers were synthesized by Sangon Company (Shanghai, China).

### 2.5. Bub3 Knockdown by RNAi

The small interfering RNAs (siRNAs) were designed and synthesized by Genepharma Company (Suzhou, China) and used to knock down the expression of Bub3 in Vero-E6 cells. The sequences were as follows: for RNAi-Bub3 sense strand, 5′-GCAGAUUUACCCAGUCAAUTT-3′, antisense strand 5′- AUUGACUGGGUAAAUCUGCTT-3′; for negative control siRNA sense strand, 5′-UUCUCCGAACGUGUCACGUTT-3′, antisense strand 5′-ACGUGACACGUUCGGAGAATT-3′. Vero-E6 cells were transfected with siRNAs using Lipofectamine 3000 reagent per the manufacturer’s instruction. Double transfections were performed at 22-h intervals. Twenty-two hours after the second transfection, cells were harvested for further analysis.

### 2.6. Yeast Two-Hybrid Screen and Colony Filter Assay

A yeast two-hybrid screen was performed according to the manufacturer’s protocol (Matchmaker Two-hybrid System 3). Briefly, chicken cDNA expression library fusion to the GAL4-activation domain in the pGADT7 plasmid was introduced by transformation into the Saccharomyces cerevisiae Y187. The bait plasmid expressing p17 fused with the GAL4-binding domain was cloned into pGBKT7, introduced by transformation into Saccharomyces cerevisiae AH109. The cDNA library clones expressing the interacting prey proteins were screened by yeast mating. Positive clones were selected on SD/-Ade/-His/-Leu/-Trp medium and tested for beta-galactosidase activity (LacZ+) by colony-lift filter assay. Here, yeast transfected with beta-galactosidase positive control (pGBKT7-p53 and pGADT7-T) or negative control (pGBKT7-Lam and pGADT7-T) as well as other yeast colonies were checked periodically for the blue color change. Each of the selected colonies was isolated and cDNAs were amplified for sequencing, then the results were subjected to a BLAST search against the NCBI database.

### 2.7. Western Blot

Vero-E6 cells were seeded on 12-well plates and cultured for 12 h before transfection with recombinant plasmids or RNAi controls or RNAi-Bub3 using Lipofectamine 3000. After transfection, cell lysates were prepared using a lysis buffer (50 mM Tris -HCl, pH 8.0, 150 mM NaCl, 5 mM EDTA, 1% NP-40, 10% glycerol, 1× complete cocktail protease inhibitor), boiled with a SDS loading buffer for 10 min, and fractionated by electrophoresis on 10% SDS-PAGE gels. The resolved proteins were then transferred onto polyvinylidene difluoride (PVDF) membranes. After blocking with 5% skimmed milk (37 °C, 1 h), the membranes were incubated with antibodies (4 °C, overnight), followed by HRP-conjugated secondary antibodies. Blots were developed using an enhanced chemiluminescence (ECL) kit (Millipore, Boston, MA, USA) per the manufacturer’s instructions. Finally, ImageJ software (version 1.52) was used for the analysis of band density.

### 2.8. Co-Immunoprecipitation (Co-IP) Assay

For co-IP, Vero-E6 cells were seeded on 6-well plates and cultured for 12 h, followed by infection with ARV S1133 or co-transfection with recombinant plasmids. At different time points after treatment, the cells were lysed with a lysis buffer on ice for 20 min. The cell lysate supernatant was collected by centrifugation and then precleared by incubation with protein A/G plus agarose for 2 h at 4 °C. The supernatant was incubated with indicated antibodies at 4 °C for 3 h and then mixed with protein A/G plus agarose and incubated for another 3 h. Beads were washed six times with a lysis buffer and boiled with a 2 × SDS loading buffer for 10 min. The samples were subjected to Western Blot analyses.

### 2.9. Confocal Laser Scanning Microscopy Assays

Vero-E6 cells were seeded on 12-well plates and cultured for 12 h, followed by infection with ARV S1133 or transfection with recombinant plasmids (1 ug per well). At different time points after treatment, the one-layer cells were fixed with 4% paraformaldehyde, permeabilized with 0.2% Triton X-100. After washing with 0.1 M phosphate-buffered saline (PBS) and blocking with 1% bovine serum albumin (BSA), the cells were successively incubated with primary antibodies for 1 h at 37 °C and with secondary antibodies for 45 min at 37 °C, followed by thrice washing as described above, then the nuclei were stained with DAPI. The samples were observed under a confocal laser scanning microscope (Olympus Corporation, Tokyo, Japan).

### 2.10. Cell-Cycle Analysis

Vero-E6 cells were seeded on 12-well plates and cultured for 12 h, followed by infection with ARV S1133 or transfection with recombinant plasmids. Different time points after treatment, cell lysates were harvested, stained with propidium (PI) using cell cycle and apoptosis analysis kits, and examined by flow cytometry (BD FACSCalibur, Piscataway, NJ, USA). The data were then analyzed with CellQuest Pro software, version 5.1 (BD).

### 2.11. Measurement of ARV Growth in Vero-E6 Cells

Vero-E6 cells receiving Bub3-specific siRNA or control siRNA were infected with ARV at an MOI of 1, and cell cultures were collected at different time points (12, 24, 36 and 48 h) post ARV infection. The samples were subjected to three rounds of freeze-thawed treatment and centrifuged at 6000× *g* for 10 min. The viral contents in supernatants or cell cultures were titrated using 50% tissue culture-infective doses (TCID_50_) in Vero-E6 cells. Cells were continuously cultured for 5 days at 37 °C in a 5% CO_2_ incubator. Tissue culture wells with cytopathic effect (CPE) were determined as positive. The titer was calculated based on a previously described method [32].

### 2.12. Statistical Analysis

All results were presentative of three independent experiments and data were presented as means ± standard deviations (SD). The statistical analysis was performed using GraphPad Prism version 9.1.1. Comparisons between groups were analyzed by Student’s *t*-test. Two-way analysis of variance (ANOVA) difference test was used to compare statistical comparisons between both treated and untreated groups. *p* ≤ 0.05 was denoted to be statistically significant (* *p* < 0.05, ** *p* < 0.01, *** *p* < 0.001).

## 3. Results

### 3.1. Ectopic Expression of ARV p17 Induces Cell-Cycle Arrest in Vero-E6 Cells

To examine if the expression of ARV p17 could prompt cell-cycle arrest, the Vero-E6 cells were transfected with plasmids expressing ARV p17 or infected with ARV, and cell-cycle progresses were monitored with flow cytometry analyses. The results showed that Flag-p17 protein was well expressed in the host cell (Figure 1A). Sigma B, a structural protein encoded by ARV S3 gene, was also examined by Western Blot assays due to its property of a major constituent of virion outer capsid presenting stably throughout the period of viral infection (Figure 1B). Furthermore, the expression of p17 was able to arrest the cell cycle of the Vero-E6 cells at the transitional phase between G2 and M, thereby preventing the progress of entering mitosis. Notably, we found that both p17 overexpression and ARV infection could retard cell proliferation, and a much higher potency was exhibited in the delayed cell cycle over time (Figure 1C,D).

### 3.2. Identification of Cellular Proteins Interacting with p17

We hypothesized that p17 interfered with the cell cycle by directly interacting with a cellular protein involved in cell-cycle modulation. Accordingly, we screened for p17 interacting proteins using a yeast two-hybrid approach. AH109 yeast cells containing pGBKT7-p17 were mated with the Y187 yeast strain containing the chicken spleen cDNA library. Then the cells were plated and selected on a quadruple dropout medium supplemented with an essential growth medium lacking tryptophan, histidine, adenine and leucine. The selected clones were confirmed by the colony PCR technique (Figure 2A). Next, the verified hits were sequenced, and the results were analyzed by the STRING analysis and demonstrated in Figure 2B and Table 1. The colony-lift assay was performed to check for the interaction between p17 and the selected proteins in yeast, and a total of three candidate proteins with cell-cycle regulatory functions (Figure 2C) turned blue, which were considered positive.

### 3.3. p17 Interacts with The Cellular Protein Bub3

From the list of candidates identified as positive hits in the yeast two-hybrid assay, we further explored the role of Bub3 in ARV infection as it was previously characterized as a checkpoint protein in mitosis [33]. A plasmid capable of expressing Myc-tagged Bub3 in mammalian cells was constructed and the interaction between p17 and Bub3 in the HEK 293T cells was verified. As shown in Figure 3A, both p17 and Bub3 were expressed in the HEK 293T cells, and Myc-Bub3 was present in lysates immunoprecipitated with anti-Flag antibodies, confirming the interaction monitored in the yeast cells. Next, the subcellular localization of overexpressed p17 and Bub3 was examined. RFP-tagged p17 and GFP-tagged Bub3 were co-expressed in the HEK 293T cells, and their localization was observed by confocal microscopy. As indicated in Figure 3B, the expression of p17 or Bub3 alone exhibited a diffuse pattern in the HEK 293T cells, but remarkably, the co-expression of p17 and Bub3 co-localized and aggregated to one side of the nucleus. To further validate the interconnection between p17 and Bub3, the interaction between overexpressed p17 and endogenous Bub3 in the Vero cells was examined using co-IP and a laser confocal microscopy. The interaction of viral p17 with endogenous Bub3 was readily detectable in the Vero-E6 cells, and viral p17 and endogenous Bub3 co-localized in the nucleus, which was somewhat different from the colocalization of exogenous proteins (Figure 3C,D). These results clearly corroborated the interaction between p17 and Bub3 in mammalian cells.

### 3.4. Bub3 Silence Alleviated Cell-Cycle Arrest Induced by ARV in Vero-E6 cells

Various studies have confirmed Bub3 as an essential component responsible for mitotic checkpoint regulation. We aimed to investigate if the interaction between p17 and Bub3 would disturb the cell cycle in the presence of ARV infection. siRNAs targeting Bub3, as well as the negative control siRNA (RNAi-Ctrl), were designed and transfected into the Vero cells, and the efficient silence of Bub3 with RNAi-Bub3–2 was verified using Western Blot (Figure 4A). To examine the effect of Bub3 on cell-cycle arrest, cells pre-transfected with RNAi-Ctrl or RNAi-Bub3 were transfected with pRK5-Flag-p17 or infected with ARV at an MOI of 1, followed by flow cytometry analysis at 24, 48 and 72 h post infection. It was found that knockdown of Bub3 slightly arrested the cell cycle, and the cell-cycle arrest caused by p17 expression was attenuated while knocking down the expression of Bub3 (Figure 4B,C). Upon ARV infection, the cell-cycle arrest in the ARV-infected cells was also alleviated by knockdown of Bub3 (Figure 4D,E). Meanwhile, the inhibited proliferation of the ARV-infected cells was significantly restored by Bub3 interference (Figure 4F). These results indicated that Bub3 played an important role in ARV-induced cell-cycle arrest.

### 3.5. Both p17 Overexpression and ARV Infection Decreased Expression of Bub3

Since Bub3 acts as a predominant effector molecule for mediating cell-cycle progression, we assumed that p17 overexpression and ARV infection might affect the Bub3 contents in the host cells. To test this hypothesis, the expression of Bub3 in the pEGFP-p17-transfected cells or the ARV-infected cells was examined by Western Blot. It was found that the overexpression of p17 markedly downregulated the Bub3 expression at different times after transfection, compared to controls (Figure 5A,B). Meanwhile, the Bub3 contents were significantly reduced in the cells post ARV infection at an MOI of 1 and 10 (Figure 5C,D). Likewise, a moderate decrease in the Bub3 levels had also been observed in the ARV-infected cells at 24 and 48 h post infection (Figure 5E,F). Moreover, to illustrate the potential factor contributing to the variation in Bub3 protein levels, we also examined the mRNA levels of Bub3 using qRT-PCR (Appendix A). The results indicated that ARV p17 would downregulate the gene levels of Bub3. Hence, the Bub3 protein levels decreased in the host cells. Altogether, these observations provide insights into the role of p17 or ARV in mediating Bub3 levels.

### 3.6. Knockdown of Bub3 Suppressed ARV Growth

Subversion of the cellular machinery that controls replication of the infected host cell is a common activity of viruses. Some of them might take advantage of disordering cell proliferation pathways for their spread [18]. As Bub3 helps to maintain normal cell-cycle progression, we proposed that Bub3 might play a critical role in the anti-ARV response of the host cells, and it would be intriguing to determine whether the deregulation of Bub3 expression contributed to the restriction of ARV replication. First, the levels of Sigma B mRNA and protein were examined by qRT-PCR and Western Blot assays in the case of knocking down the expression of Bub3. The results showed that the knockdown of Bub3 in the Vero-E6 cells led to an attenuated Sigma B expression at both mRNA and protein levels (Figure 6A,D). To consolidate this finding, viral loads of ARV in the cell cultures or the supernatants were monitored at different time points post infection (12 h, 24 h, 36 h and 48 h) in the Bub3 knockdown or control cells. As shown in Figure 6E,F, the knockdown of Bub3 significantly reduced ARV titers compared to the controls in both cell cultures and the supernatants, suggesting that the Bub3-mediated cell-cycle arrest might be a strategy employed by ARV to support viral propagation.

Taken together, our data showed that ARV p17 triggered the cell-cycle perturbation via interacting with and downregulating the expression level of cellular protein Bub3 to alter a favorable environment for effective replication, which might shed new light on the dynamic interplay between avian reoviruses and host cell-cycle machinery.

## 4. Discussion

ARV is the causative agent of RSS in chickens and turkeys. Clinical signs of the disease include retarded growth and poor weight gain. Infection of ARV leads to considerable losses to the commercial poultry industry worldwide, primarily in broilers. One of the reovirus proteins, the ARV p17, was found to function as the cell proliferation rate retardant without causing apoptosis [16]. Although there are some reports on the involvement of ARV p17 in signaling pathways to modulate host cell cycle [15,34], the molecular mechanism underlying ARV-induced growth inhibition has not been well described. In this study, we first identified several cellular proteins capable of interacting with p17 in the host cells by a yeast two-hybrid assay. Among these cellular proteins, the mitotic checkpoint protein Bub3 attracted our attention, as Bub3 has been identified to control chromosome segregation and monitor kinetochore-microtubule interactions. After that, the alleviation of mitotic arrest in the G2/M phase caused by the p17 expression or the ARV infection was observed in the cells with a knockdown of the Bub3 expression (Figure 4). Furthermore, both the p17 overexpression and the ARV infection downregulated the expression level of Bub3 (Figure 5). We also overexpressed Bub3 in these cells to detect its effects; however, there seemed to be no significant statistical differences due to the high background value of Bub3 in the host cells.

Most viral proteins or compounds are commonly used to block the cell cycle in the G2/M phase by destabilization of microtubules, which then inhibits spindle dynamics, thereby leading to mitotic arrest. Of note, even though there are few published studies on Bub3, it is one of the core members of SAC to ensure the fidelity of chromosome segregation, monitoring the normal cell-cycle progression. Evidence for the existence of Bub3 functions in mitosis is provided by studies reporting that the altered Bub3 expression levels would significantly impair the mitotic checkpoint mechanism, and the cells lacking Bub3 also have an additional role in delaying the anaphase onset [28,35,36,37]. In our study, the overexpression of or the reduction in the Bub3 level did not seem to significantly alter the cell-cycle progression, as shown in Figure 4C. However, contradictory reports have verified that the absence of Bub3 would not overtly impair SAC stability [33,38,39,40], which may explain why Bub3 could only partially drive the cell cycle in this paper. Nevertheless, our subsequent finding showed that Bub3 depletion would facilitate mitotic progression accompanied by p17 overexpression or ARV infection, indicating the unique role of Bub3 in regulating cell growth in the case of ARV infection.

The dysregulated expression of Bub3 has been proven to be associated with various malignancies [38], but no other studies have been conducted to demonstrate the role of Bub3 in other cellular biological functions. This is the first report demonstrating the critical role of Bub3 in host response to avian viruses. In this study, it was found that targeting Bub3 by ARV p17 might be a common strategy to manipulate the cell-cycle progression during ARV infection, and the interconnection of Bub3 and p17 may also affect the formation of progeny virions. Considering that some cell-cycle regulators can influence both cell division and programmed cell death according to different cellular environments and genetic backgrounds [41], Bub3 might also play an important but not a sole role in the different stages of viral infection.

ARV p17 can shuttle continuously between the nucleus and the cytoplasm, subsequently accumulating in the nucleoplasm of the infected or transfected cells [42]. It was reported that p17 expression retarded the growth of the cells through the activation of p53 and p21^cip1/waf1^ [16]. In addition, p17 interacts with several cyclins in the cytoplasm and possesses broad inhibitory effects on cell-cycle CDKs (cyclin-dependent kinases), as well as the CDK-cyclin complexes in various cancer cell lines [34]. Suppression of both CDK1 and Plk1 functions by p17 disrupts the phosphorylation of vimentin and causes G2/M cell-cycle arrest, and thus benefits virus replication [30]. In the present study, we reported an interaction between ARV p17 and Bub3, and this interaction constitutes the critical role played by p17 involved in cell-cycle signaling. However, further research is still needed to determine whether ARV p17 induces inhibition of cell proliferation via other pathways. In the meantime, the consequences of cell-cycle abnormality for the virus life cycle are not well defined. Thus, the extent to which the perturbation of the G2/M transition is involved in ARV infection remains to be further investigated.

In conclusion, the host Bub3 was found to interact with ARV p17 upon viral infection, which was important for arresting the host cell cycle in the G2/M phase. The decrease in Bub3 expression efficiently mitigated the cell-cycle arrest and inhibited viral growth. Our results provided new insights into the mechanism underlying the ARV-manipulated cell cycle via hijacking host mitotic protein Bub3 during infection.

## Figures and Tables

**Figure 1 viruses-14-02385-f001:**
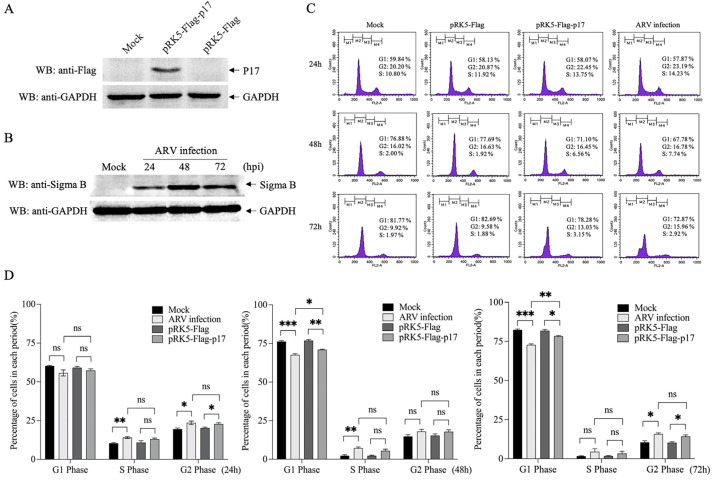
Examination of the effects of overexpressed Flag-p17 and ARV infection on Vero-E6 cell cycle. (**A**) Expression of p17 protein in Vero-E6 cells. Vero-E6 cells were seeded on 12-well plates and cultured overnight. Cells were then transfected with pRK5-Flag or pRK5-Flag-p17 plasmid. Twenty-four hours after transfection, cell lysates were prepared and examined with anti-Flag and anti-GAPDH antibodies. (**B**) Examination of ARV infection. Vero-E6 cells were infected with ARV at an MOI of 1. Cell lysates were extracted at different time points (24 h, 48 h and 72 h) post infection and examined by Western Blot using anti-Sigma B and anti-GAPDH antibodies. (**C**) p17 overexpression or ARV infection induced cell-cycle arrest at G2/M phase. Vero-E6 cells were treated with pRK5-Flag-p17 transfection or ARV infection (MOI = 1). At different time points (24 h, 48 h and 72 h), cells remaining in different phases were analyzed by flow cytometry. (**D**) The percentage of all period cells in (**C**) was quantified. * stands for *p* < 0.05, ** stands for *p* < 0.01 and *** stands for *p* < 0.001.

**Figure 2 viruses-14-02385-f002:**
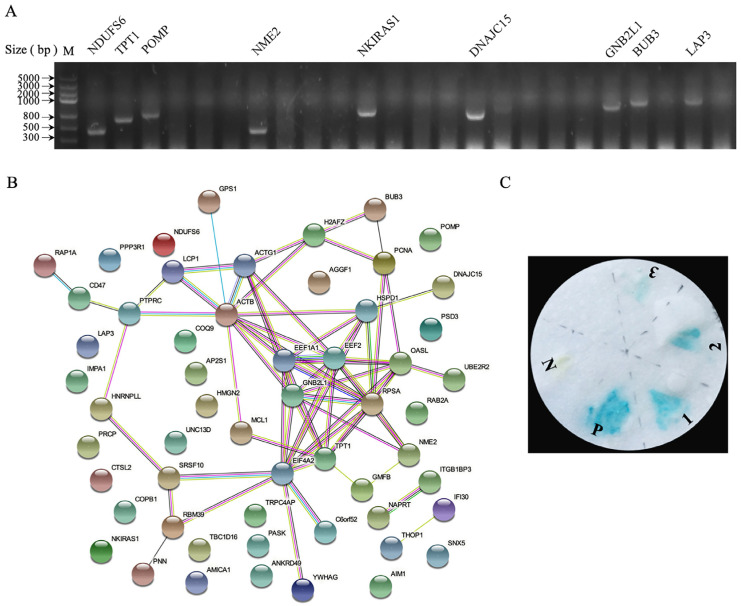
Yeast two-hybrid screen of p17 interacting proteins. (**A**) Confirmation of positive clones by PCR. (**B**) PPI network analysis. The network of cellular proteins interacting with ARV p17 was constructed using STRING 11.5. (**C**) The colony-lift assay of p17 and selected proteins. N, negative control; P, positive control; 1, eEF2; 2, TPT1; 3, Bub3.

**Figure 3 viruses-14-02385-f003:**
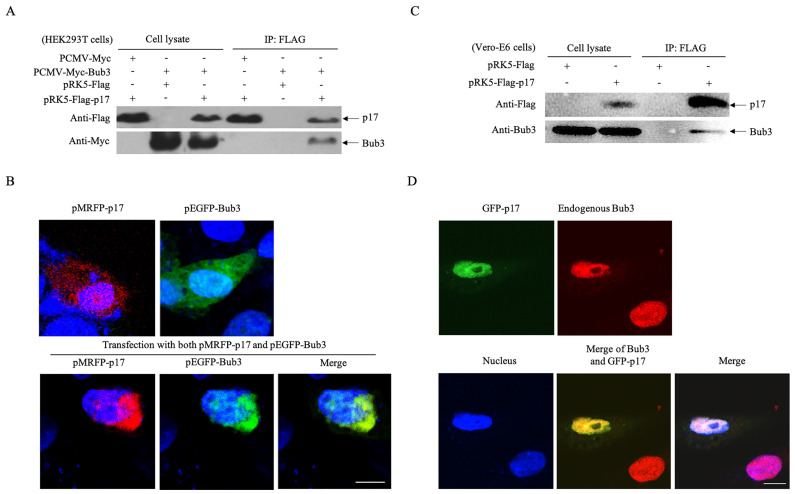
Interaction of ARV p17 with cellular protein Bub3. (**A**) Confirmation of the interaction of p17 with exogenous Bub3 in HEK 293T cells. Cells were transfected with the indicated plasmids. Twenty-four hours after transfection, cell lysates were prepared and immunoprecipitated with an anti-Flag antibody and immunoblotted with an anti-Flag or anti-Myc antibody. (**B**) Colocalization of exogenous p17 and Bub3 in HEK 293T cells. Cells were co-transfected with pMRFP-p17 (red) and pEGFP-Bub3 (green). Twenty-four hours after transfection, cells were fixed with 4% paraformaldehyde and the cell nuclei was counterstained with DAPI (blue). The cell samples were observed with a confocal laser scanning microscope (magnification, ×600). The scale bar in the picture represents 5 μm. (**C**) Interaction of p17 with endogenous Bub3 in Vero-E6 cells. Cells were transfected with the indicated plasmids. Twenty-four hours after transfection, cell lysates were prepared and immunoprecipitated with an anti-Flag antibody and immunoblotted with an anti-Flag or anti-Bub3 antibody. (**D**) Colocalization of p17 with endogenous Bub3 in Vero-E6 cells. Cells were transfected with pEGFP-p17 plasmids. Twenty-four hours after transfection, cells were fixed with 4% paraformaldehyde and permeabilized with 0.2% Triton X-100. Then the cells were stained with a rabbit anti-Bub3 monoclonal antibody, followed by a TRITC-conjugated goat anti-rabbit antibody (red). Nuclei was counterstained with DAPI (blue). The cell samples were observed with a confocal laser scanning microscope (magnification, ×600). The scale bar in the picture represents 10 μm.

**Figure 4 viruses-14-02385-f004:**
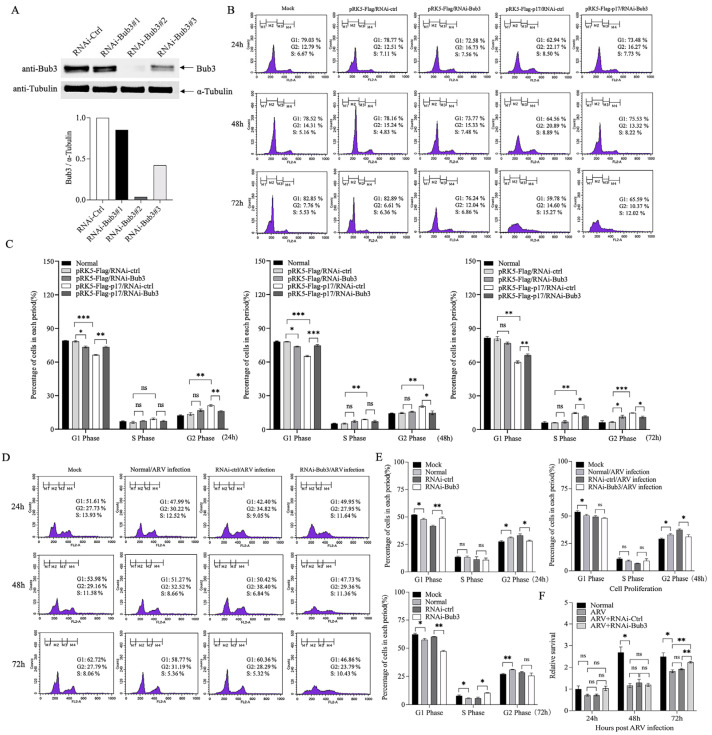
Bub3 is required for ARV-induced cell-cycle arrest. (**A**) Effects of Bub3 RNAi on the expression of endogenous Bub3. Vero-E6 cells were transfected with siRNA (RNAi#1–3) or controls. Twenty-four hours after transfection, cell lysates were prepared and examined by Western Blot using an anti-Bub3 antibody. Endogenous α-Tubulin expression was examined as an internal control, and the relative levels of Bub3 were calculated as follows: band density of Bub3/that of α-Tubulin. (**B**) Knockdown of Bub3 attenuated cell-cycle arrest in cells with overexpression of p17. Vero-E6 cells were transfected with RNAi-Bub3 or controls, followed by transfection with pRK5-Flag-p17 or empty vector. At different time points (24 h, 48 h and 72 h) after p17 overexpression, cell-cycle changes were analyzed by flow cytometry. (**C**) The percentage of all period cells in (**B**) was quantified. * stands for *p* < 0.05, ** stands for *p* < 0.01 and *** stands for *p* < 0.001. (**D**) Knockdown of Bub3 attenuated cell-cycle arrest in ARV-infected cells. Vero-E6 cells were transfected with siRNA or controls, followed by infection with ARV at an MOI of 1. At different time points (24 h, 48 h and 72 h) after ARV infection, cell-cycle changes were analyzed by flow cytometry. (**E**) The percentage of all period cells in (**D**) was quantified. * stands for *p* < 0.05, ** stands for *p* < 0.01 and *** stands for *p* < 0.001. (**F**) Knockdown of Bub3 restored the cell proliferation inhibited by ARV infection. Vero-E6 cells were transfected with siRNA or controls, followed by infection with ARV at an MOI of 1. At different time points (24 h, 48 h and 72 h) post ARV infection, the cell proliferation was analyzed with MTT. * stands for *p* < 0.05, ** stands for *p* < 0.01 and *** stands for *p* < 0.001.

**Figure 5 viruses-14-02385-f005:**
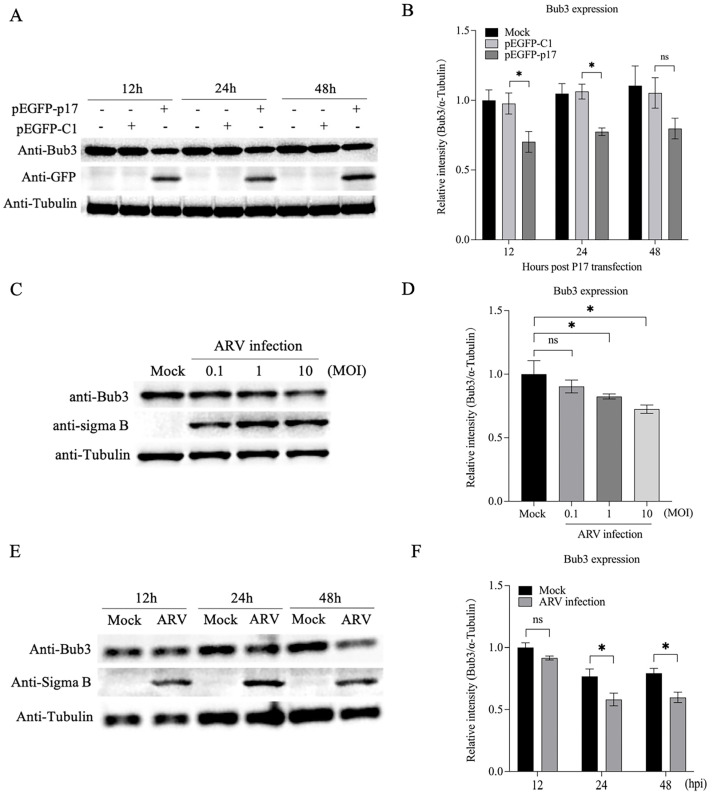
Both p17 overexpression and ARV infection result in a reduction in Bub3 in cells. (**A**,**B**) Overexpression of p17 decreased the Bub3 protein level. Vero-E6 cells were transfected with pEGFP-p17 or empty vectors, and cell lysates were examined by Western Blot using anti-GFP and anti-Bub3 antibodies at different time points (12 h, 24 h and 48 h) after transfection. The levels of Bub3 were normalized to that of α-Tubulin. (**C**,**D**) ARV infection suppressed the expression of endogenous Bub3. Vero-E6 cells were infected with ARV at different MOIs (0.1, 1 and 10). Twenty-four hours after infection, cell lysates were extracted and examined by Western Blot, and the levels of Bub3 were normalized to that of α-Tubulin. (**E**,**F**) Vero-E6 cells were infected with ARV at an MOI of 1. Cell lysates were extracted at different time points (12 h, 24 h and 48 h) post infection and examined by Western Blot. The levels of Bub3 were normalized to that of α-Tubulin. * stands for *p* < 0.05.

**Figure 6 viruses-14-02385-f006:**
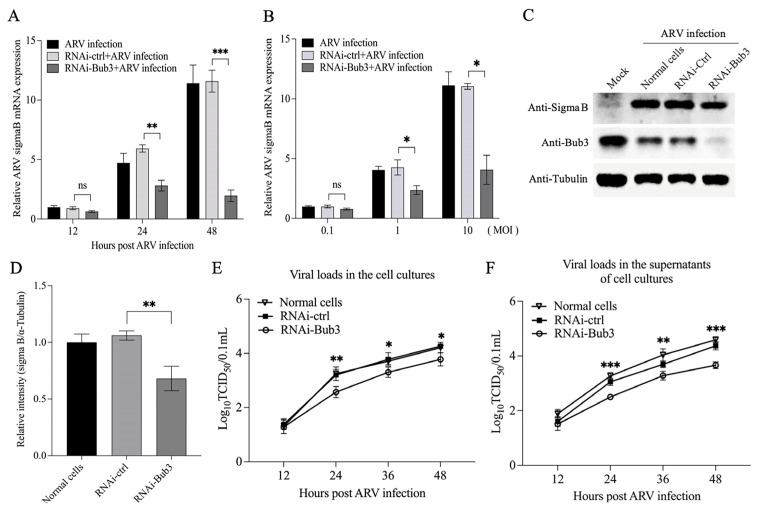
Effects of Bub3 on ARV growth in Vero-E6 cells. (**A**) Vero-E6 cells were transfected with RNAi-Bub3 or controls, followed by infection with ARV at an MOI of 1. At different time points (12 h, 24 h, and 48 h) post infection, the level of Sigma B mRNA was quantified. The relative level of Sigma B was calculated as follows: Sigma B expression/that of GAPDH expression. (**B**) Vero-E6 cells were transfected with RNAi-Bub3 or controls, followed by infection with ARV at an increased MOI of 0.1, 1 and 10. The level of Sigma B mRNA were quantified after twenty-four hours of ARV infection. ** stands for *p* < 0.01 and *** stands for *p* < 0.001. (**C**,**D**) Vero-E6 cells were transfected with RNAi-Bub3 or controls, followed by infection with ARV at an MOI of 10. Cell lysates were prepared at 24 h post infection and examined with Western Blot using indicated antibodies, and the band densities of Sigma B in (**C**) were quantitated by densitometry (**D**). Endogenous α-Tubulin expression was examined as an internal control. (**E**,**F**) Knockdown of Bub3 inhibited ARV replication. Vero-E6 cells were treated with RNAi to knock down endogenous Bub3 expression, followed by infection with ARV at an MOI of 1. At different time points (12 h, 24 h, 36 h and 48 h) post ARV infection, the viral loads in the cell cultures (**E**) or supernatants (**F**) were determined by TCID_50_ analysis using 96-well plates. * stands for *p* < 0.05, ** stands for *p* < 0.01 and *** stands for *p* < 0.001.

**Table 1 viruses-14-02385-t001:** The screening result of chicken spleen cDNA using ARV p17.

Protein Name	No. GenBank	Function
Induced myeloid leukemia cell differentiation protein (MCL1)	XP_040508245.1	Regulation of cell differentiation
Leucine aminopeptidase 3 (LAP3)	NP_001026507.1	Peptidase activity
Tumor protein, translationally-controlled 1 (TPT1)	NP_990729.1	Regulation of intrinsic apoptotic signaling pathway in response to DNA damage
NF-kappa-B inhibitor-interacting Ras-like protein 1 (NKIRAS1)	XP_046794149.1	Regulation of I-kappaB kinase/NF-kappaB signaling
Eukaryotic translation elongation factor 1 alpha 1 (eEF1A1)	NP_001308445.1	Translation elongation factor activity, GTPase activity
Guanine nucleotide-binding protein beta subunit2-like 1 (GNB2L1)	BAJ53010.1	RNA binding, regulation of protein kinase activity
Eukaryotic translation elongation factor 2 (eEF2)	NP_990699.2	Regulation of cell growth
Mitotic checkpoint protein (Bub3)	NP_001006506.1	Ubiquitin binding, regulation of mitotic cell cycle
Eukaryotic initiation factor 4A-II (EIF4A2)	NP_989880.1	Translation initiation factor activity, RNA helicase activity
2′-5′-oligoadenylate synthase-like protein (OASL)	XP_046782048.1	Regulation of viral genome replication and RIG-I pathway
Ubiquitin-conjugating enzyme E2 R2 (UBE2R2)	NP_001026582.2	Ubiquitin-conjugating enzyme activity
Heat shock protein family D member 1 (HSPD1)	NP_001012934.1	Ubiquitin protein ligase binding
Nucleoside diphosphate kinase (NME2)	NP_990378.1	Regulation of apoptotic process
Serine/arginine-rich splicing factor 10 (SRSF10)	XP_015153161.1	Regulation of mRNA splicing, cytosolic transport
DnaJ homolog subfamily C member 15 (DNAJC15)	NP_001264705.2	ATPase activator activity, regulation of mitochondrial electron transport

## Data Availability

Not applicable.

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
