# Peer review of "Arrest of Cell Cycle by Avian Reovirus p17 through Its Interaction with Bub3"

_viruses, 2022, doi:10.3390/v14112385_

Round 1
Reviewer 1 Report
This article unfolds through the discovery that the P17 protein of ARV can influence the cell cycle. It also identifies Bub3, a protein that interacts with the P17 protein, and demonstrates through overexpression and RNA interference assays that Bub3 is associated with delayed cell cycle, and finally demonstrates that Bub3 also has an effect on viral replication. The overall logic is clear and complete, but there are still a few questions to be addressed below.
Major comments
Line 198: In Figure C, MOCK, ARV infection, pRK5-FLAG, and pRK5-fLAG-P17 data sets should be examined in a single cousin. Is there a distinction between these two because ARS infection and protein p17 can both induce cell cycle delays?
Line 307: Figures A to F depict the impact on Bub3 protein levels; is there evidence to support a gene-level impact? Because potential degradation and a decrease in gene levels are both contributing factors to the decline in protein levels?
Minor comments
Line 198: The authors should have included a picture of a western bolt demonstrating ASV infection to support this. The authors should have included a picture of a western bolt proving ASV infection to support this.
Reviewer 2 Report
The authors found that ARV p17 arrested the host cell cycle in G2/M phase. They then identified host proteins that interact with ARV p17 and further focused on the interaction between ARV p17 and host mitotic protein Bub3. Hijacking Bub3 during ARV infection was proposed as a mechanism underlying ARV-manipulated cell cycle. The study is interesting and the data are sound.
I have only one comment about the results. It will be helpful if a gel picture of p17 expression in ARV infected cells can be provided in Figure 1A.
However I did find many places where English writing can be improved. The following is only a few examples:
Line 33: ARV should be spelled out. It is the first time it is used in the body of the paper.
Line 37: add comma before “including”. “Large” should be changed to “large”. Otherwise change “median” and “small” to “Median” and “Small”, respectively.
Line 45: In the sentence “The eukaryotic cell cycle is typically separated into 4 phases: G1, S, G2 and M. “, the four phases should be spelled out, such as “the S or synthesis phase, the M or mitosis phase, G1 and G2, the gap phases”. The information can be included either within the same sentence, or in the text afterwards, where the four phases were described in details.
Line 67: “evolved” should be “have evolved”.
Line 69: “Examples of which” can be shorten to “Examples”.
LIne 70: “Recent study exhibited” should be “Recent studies showed”.
Line 71: “expression of ARV p17 resulted in rapid cell-cycle arrest [31, 16, 17], unfortunately,” should be “expression of ARV p17 resulted in rapid cell-cycle arrest [31, 16, 17]. Unfortunately,”
Lines 73 and 74: The sentence is very long. It could be divided into two shorter ones, such as the following: "In this study, we aimed to explore strategies employed by p17 to retard cell proliferation. We discovered a new approach, by which p17 functions to hamper cell cycle progression."
Line 237: “It was been” should be “it has been”.
Fig 2A legend is difficult to read. It could be replaced by something simpler such as “Confirmation of positive clones by PCR. “
...
